# Ribosome profiling reveals downregulation of UMP biosynthesis as the major early response to phage infection

Patrick B. F. O'Connor,[1,2] Jennifer Mahony,[3] Eoghan Casey,[3] Pavel V. Baranov,[1] Douwe van Sinderen,[3] Martina M. Yordanova[1]

**ABSTRACT** Bacteria have evolved diverse defense mechanisms to counter bacteriophage attacks. Genetic programs activated upon infection characterize phage–host molecular interactions and ultimately determine the outcome of the infection. In this study, we applied ribosome profiling to monitor protein synthesis during the early stages of sk1 bacteriophage infection in *Lactococcus cremoris*. Our analysis revealed major changes in gene expression within 5 minutes of sk1 infection. Notably, we observed a specific and severe downregulation of several *pyr* operons which encode enzymes required for uridine monophosphate biosynthesis. Consistent with previous findings, this is likely an attempt of the host to starve the phage of nucleotides it requires for propagation. We also observed a gene expression response that we expect to benefit the phage. This included the upregulation of 40 ribosome proteins that likely increased the host's translational capacity, concurrent with a downregulation of genes that promote translational fidelity (*lepA* and *raiA*). In addition to the characterization of host–phage gene expression responses, the obtained ribosome profiling data enabled us to identify two putative recoding events as well as dozens of loci currently annotated as pseudogenes that are actively translated. Furthermore, our study elucidated alterations in the dynamics of the translation process, as indicated by time-dependent changes in the metagene profile, suggesting global shifts in translation rates upon infection. Additionally, we observed consistent modifications in the ribosome profiles of individual genes, which were apparent as early as 2 minutes post-infection. The study emphasizes our ability to capture rapid alterations of gene expression during phage infection through ribosome profiling.

**IMPORTANCE** The ribosome profiling technology has provided invaluable insights for understanding cellular translation and eukaryotic viral infections. However, its potential for investigating host–phage interactions remains largely untapped. Here, we applied ribosome profiling to *Lactococcus cremoris* cultures infected with sk1, a major infectious agent in dairy fermentation processes. This revealed a profound downregulation of genes involved in pyrimidine nucleotide synthesis at an early stage of phage infection, suggesting an anti-phage program aimed at restricting nucleotide availability and, consequently, phage propagation. This is consistent with recent findings and contributes to our growing appreciation for the role of nucleotide limitation as an anti-viral strategy. In addition to capturing rapid alterations in gene expression levels, we identified translation occurring outside annotated regions, as well as signatures of non-standard translation mechanisms. The gene profiles revealed specific changes in ribosomal densities upon infection, reflecting alterations in the dynamics of the translation process.

**KEYWORDS** ribosome profiling, mRNA translation, anti-phage response

Address correspondence to Martina M. Yordanova, martina.yordanova@ucc.ie, or Douwe van Sinderen, d.vansinderen@ucc.ie.

P.V.B. is co-founder and shareholder of EIRNA Bio.

See the funding table on p. 14.

Bacteria have evolved various strategies to combat bacteriophage attacks, including phage adsorption and nucleic acid entry prevention, nucleic acid interference, and

abortive infection where cells self-sacrifice before completion of the replication cycle (1–4). Understanding the gene expression programs activated upon infection is crucial in characterizing the complex molecular phage–host interactions that ultimately determine the infection outcome. Additionally, this knowledge can inform the design of phage-resistant bacterial strains as well as facilitate the use of phages as antibacterial agents (5–7).

*Lactococcus cremoris*, a Gram-positive mesophilic bacterium, has an approximate genome size of 2.5 Mbp with a low GC content (~35%) and ~2300 annotated proteins. It has an unusually high fraction (~13%) of leaderless mRNAs (8). Belonging to the Lactobacillales order, *L. cremoris* is capable of metabolizing lactose into lactic acid, and it is utilized in starter cultures for dairy fermentation processes. Due to its significance in the dairy food industry, lactococcal phage defense mechanisms have been extensively studied, with at least 20 abortive phage infection systems (Abi) described (9).

One of the major infectious agents in commercial dairy environments is sk1, a small isometric-headed lytic phage belonging to the skunaviruses (10, 11), previously termed the 936 phage group. Sk1 has a linear double-stranded DNA genome with 54 annotated genes, divided into three classes based on their temporal expression during infection. The early genes [open reading frame (ORF) 21 to ORF50] are encoded in the negative strand, while the middle (ORF51–ORF54) and late genes (ORF1–ORF20) are encoded in the positive strand. Early, middle, and late genes were first transcribed at approximately 4, 8, and 15 minutes post-infection (p.i.), respectively (12).

Phages rely on host cell machinery, including the protein synthesis apparatus, to express and propagate their genomes. Over the past decade, Ribosome profiling, also known as Ribo-seq, has emerged as a prominent approach for studying protein synthesis in both eukaryotes and prokaryotes (13–17). High-throughput sequencing of RNA footprints, shielded from RNase digestion by translating ribosomes, unveils their positions, offering insights into translated regions of the transcriptome. Quantitative analysis of these footprints mapped to different genes enables the estimation of relative protein synthesis rates (14, 18).

The application of Ribo-seq to various Gram-positive and Gram-negative bacteria has significantly enhanced efforts to improve the accuracy of genome annotations (19). Variations in the protocol allow for the preferential capture of elongating or initiating ribosomes (20), and combining these approaches facilitates the discovery of previously unidentified translated ORFs and the prediction of novel proteoforms (21, 22). Furthermore, Ribo-seq has provided insights into the dynamics of the translation process, revealing interactions between translating ribosomes and mRNA (23, 24) instances of ribosome stalling at specific mRNA positions (16, 25, 26), as well as nonstandard translation mechanisms (27).

Given that Ribo-seq has established itself as a powerful tool for studying mRNA translation across various bacterial species and under diverse conditions, it holds significant promise for investigating gene expression changes in phage–host interactions. Notably, Ribo-seq provides more accurate measurements of instantaneous protein synthesis rates compared to traditional methods such as RNA-seq or proteomic-based approaches, which have been commonly used to study phage–host interactions thus far (28–31).

Ribo-seq has been extensively employed to comprehend the dynamics of viral–host interactions in eukaryotic systems (32–34). However, we are aware of a single study by Liu et al. (35) that reports the use of Ribo-seq to investigate the translational response during phage infection of a prokaryote (the infection of *Escherichia coli* by bacteriophage lambda). While they report mild repression of approximately 1000 genes within 20 minutes of infection induction, it failed to detect a specific response, perhaps because of numerous secondary effects occurring by this time point.

In the current study, we employed ribosome profiling to investigate the *L. cremoris* gene expression response at an earlier stage of sk1 infection. To this end, we conducted a time-series assay based on previously elucidated temporal regulation of sk1 phage

transcripts (12). Our study revealed a very specific change in gene expression occurring within the first 5 minutes of infection, revealing that Ribo-seq can be successfully employed to pinpoint immediate response behind the complex gene expression programs underlying phage–host interactions.

## RESULTS

### Ribosome profiling in sk1 infected *L. cremoris*

We performed ribosome profiling on *L. cremoris* NZ9000 infected with sk1 at 2, 5, and 15 minutes p.i. (Fig. 1A) to investigate the gene expression response during early sk1 infection. Sequencing libraries were aligned to the combined reference genomes of *L. cremoris* and sk1. Approximately, 3 million reads were mapped unambiguously for each sample (Table S1). The quantification of gene expression was, in general, highly reproducible (Fig. 1B; Fig. S1). The length of Ribo-seq reads was shorter than previously reported for other bacteria (26), i.e., ~80% of reads were within 16–26 nucleotides in length (Fig. 1C). As expected, the mapped Ribo-seq data exhibited the triplet periodicity signal produced as a result of the codon-wise movement of ribosomes (23). This periodicity signal was absent in the RNA-seq data (Fig. S2). The metagene profile produced exclusively with unambiguously mapped reads (i.e., those that mapped to one location in the genome) matched well with the characteristic Ribo-seq data: increased read density in the coding sequence (CDS) region in comparison to the adjacent, presumed noncoding regions (Fig. 1D).

The Ribo-seq alignments to sk1 revealed the translation of a novel ORF (Fig. S3). This ORF was predicted as a hypothetical protein in an alternative annotation of sk1. It is located 5′ of ORF30 and encodes a 36-residue peptide of unknown function. A BlastX query of the sequence returns a high-sequence similarity to a predicted membrane protein in *Lactococcus* phage RH6.

### Reprogramming of gene expression response occurs predominantly within 5 minutes p.i.

As may be expected, the expression of sk1 genes was found to increase with time. It nonetheless remained insubstantial relative to that of the host with the percentage of footprints aligning to sk1 genes increasing from 0.6% at 2 minutes p.i. to 1.7% at 15 minutes p.i. The number of detected differentially expressed (DE) genes (host + phage) increased with the infection progression from 215 at 2 minutes p.i., to 648 at 5 minutes p.i., and to 811 at 15 minutes p.i. (Fig. 2A).

A comparison of the DE test statistics (metric of the likelihood that a gene is DE), however, showed that these numbers alone, do not reveal the true extent of the regulation during the first 2 minutes. The remarkably strong similarity of gene expression changes across time points, even in the comparison of the test statistic obtained for the 2- and 15-minute infection time points (Fig. 2B), indicates that a major gene expression shift is observable within 2 minutes p.i. but fails to pass the statistical test threshold to be classified as DE. It passes the threshold with the amplification of the change at later time points similar to what we previously observed during other rapid translational responses (36).

Thus, the gene expression response consists of a general shift which is well underway by 2 minutes p.i. and essentially stable by 5 minutes p.i. The regulation observed after 5 minutes consists of maintaining the status quo though with a further increase of the response's magnitude (Fig. S4).

Hierarchical clustering of the gene expression profiles of sk1 at all three time points revealed one distinct clade with relatively low intra-clade variation (Fig. 2C). This was found to encapsulate all the early genes and no others. While all but two of the genes in this clade were found to be expressed at 2 minutes p.i., only two middle genes (ORF51 and ORF54) and none of the late genes were found to be DE at the 15-minute time point. It seems that, in our experiments, the temporal regulation is slower than

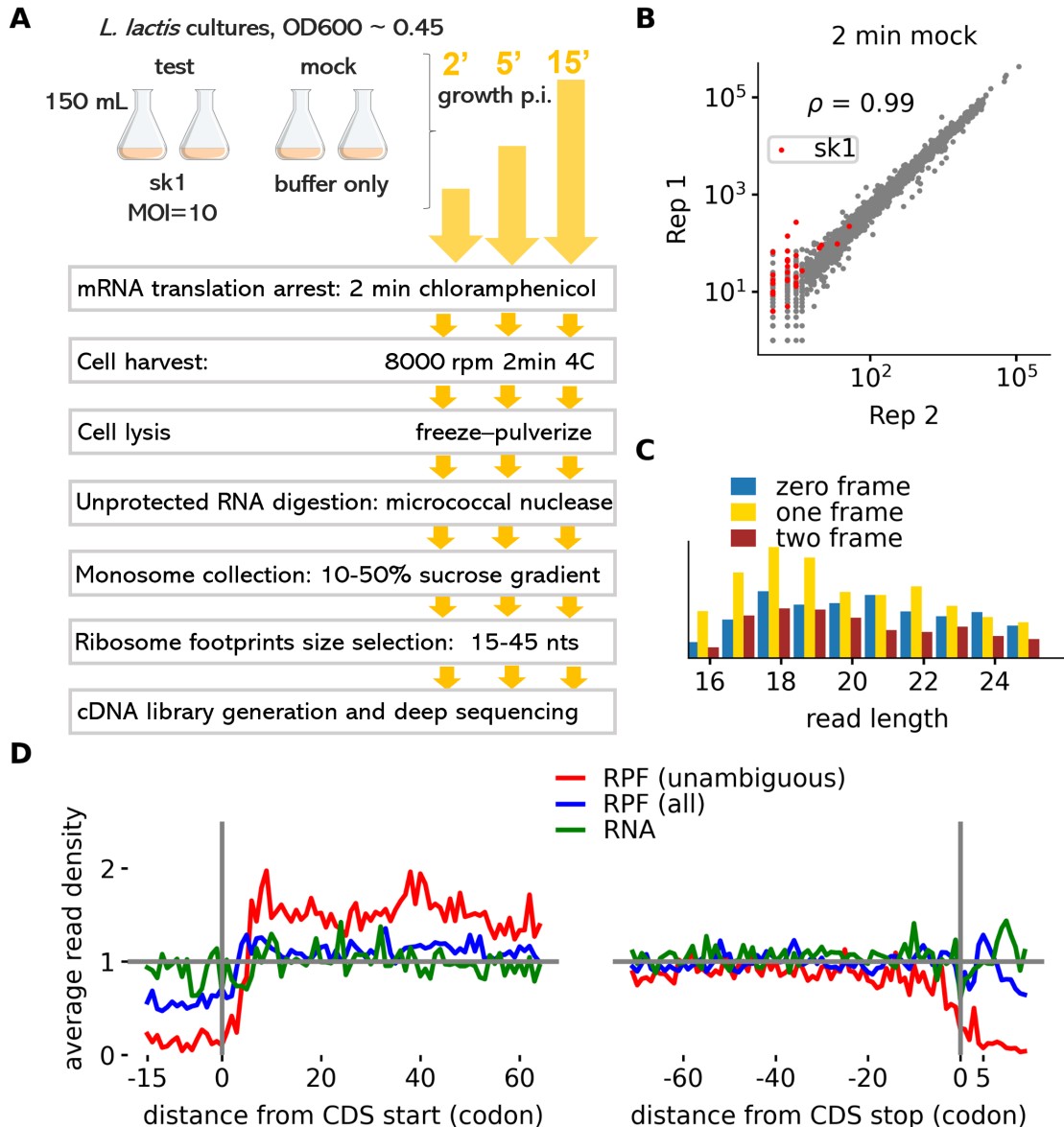

**FIG 1** Ribosome profiling of sk1 infected *L. cremoris*. (A) Experimental design. *L. cremoris* cultures were grown to an OD600 ~0.45 and inoculated with sk1 phage or buffer only. For each time interval, two sk1- and two mock-infected cultures were subjected to ribosome profiling. (B) Correlation of the number of reads aligned to each gene between two replicates. Red dots indicate sk1 genes. (C) Sub-codon phasing of mapped Ribo-seq reads is shown for each individual read length, the absolute counts are used. (D) Metagene plots at the CDS start and stop codons produced with 3′ ends of ribosome footprints. Unambiguous footprint reads (in red) are those which mapped only to a single position in the genome, while "all" (in blue) includes those reads that are mapped to multiple positions. RNA-seq reads are in green.

previously reported, but nevertheless, our experiments show strong agreement with previous findings (12).

## Severe downregulation of genes involved in UMP biosynthesis

We identified a specific small group of host genes that undergo a high fold change regulation at 5 and 15 minutes which is not observed at 2 minutes (Fig. 2B). We found that these genes are all required for the *de novo* synthesis of uridine monophosphate (UMP). The UMP biosynthesis pathway consists of six enzymatic steps leading to the formation of UMP, which may then be further converted into UTP, CTP, dCTP, or dTTP (Fig. 3). The genes encoding the corresponding enzymes are arranged in several

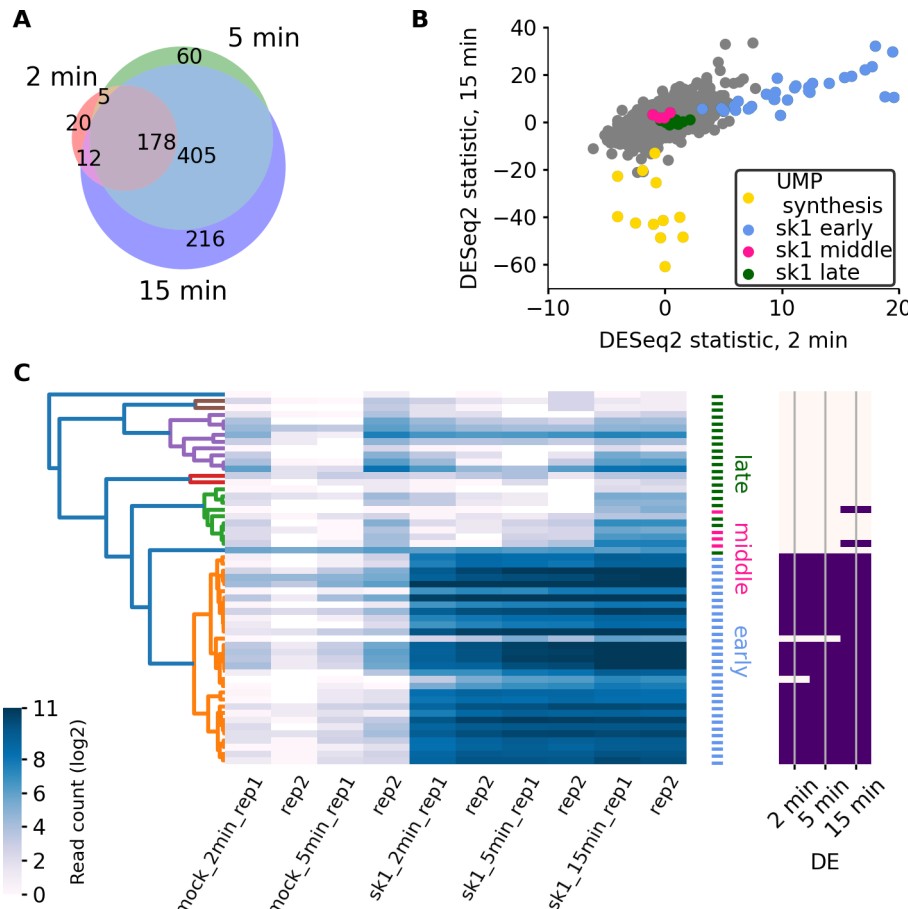

**FIG 2** The characterization of gene expression response to infection. (A) Genes identified as DE at different time points p.i. (B) Comparison of DESeq2 test statistic at 2 and 15 minutes p.i. The test statistic equates to the number of SDs from null. (C) Heatmap and hierarchical clustering of sk1 gene expression. Genes previously classified as early (indicated on the right in blue) are found to belong to one clade shown in orange in the dendrogram. The genes identified at various time points to be DE are indicated on the right.

operons: *pyrKDbFOrfC*, *carB*, *pyrH*, *pyrEC*, *pyrRPBcarA*, and *pyrDa*. Here, we found that the repression of these genes by the 15-minute time point ranges from a twofold repression for *pyrDa* to 36-fold for *pyrB* (Fig. 3).

The concerted suppression of *pyr* operon expression observed here suggests a role for the PyrR-mediated transcription regulation mechanism as a central mediator of the response. This mechanism modulates the expression in response to UMP levels and is found in many Gram-positive bacteria (37–39). PyrR is an mRNA-binding protein and with elevated UMP levels, and PyrR-UMP complex binds *pyr* mRNA leaders and promotes the formation of a terminator structure blocking transcription. The formation of an antiterminator structure at low UMP levels allows transcription to continue. As PyrR regulates the transcription of operons, the significant repression that we observe would require these mRNAs to have a very short mRNA half-life of approximately less than 2 minutes. This is close to the lower end of estimates of mRNA half-life in bacteria between 1 and 40 minutes (40–42).

The Ribo-seq data enabled us to confirm the expression of an additional gene (OrfC) in the *pyrKDbFOrfC* operon. The gene was identified to belong to this operon and was named *OrfC* (43). It is currently annotated as a hypothetical protein encoded immediately downstream of *pyrF* with locus tag LLNZ_RS05640 (Fig. S5). Like the other genes in the operon, it was found to undergo a 14-fold repression supportive of a role in pyrimidine

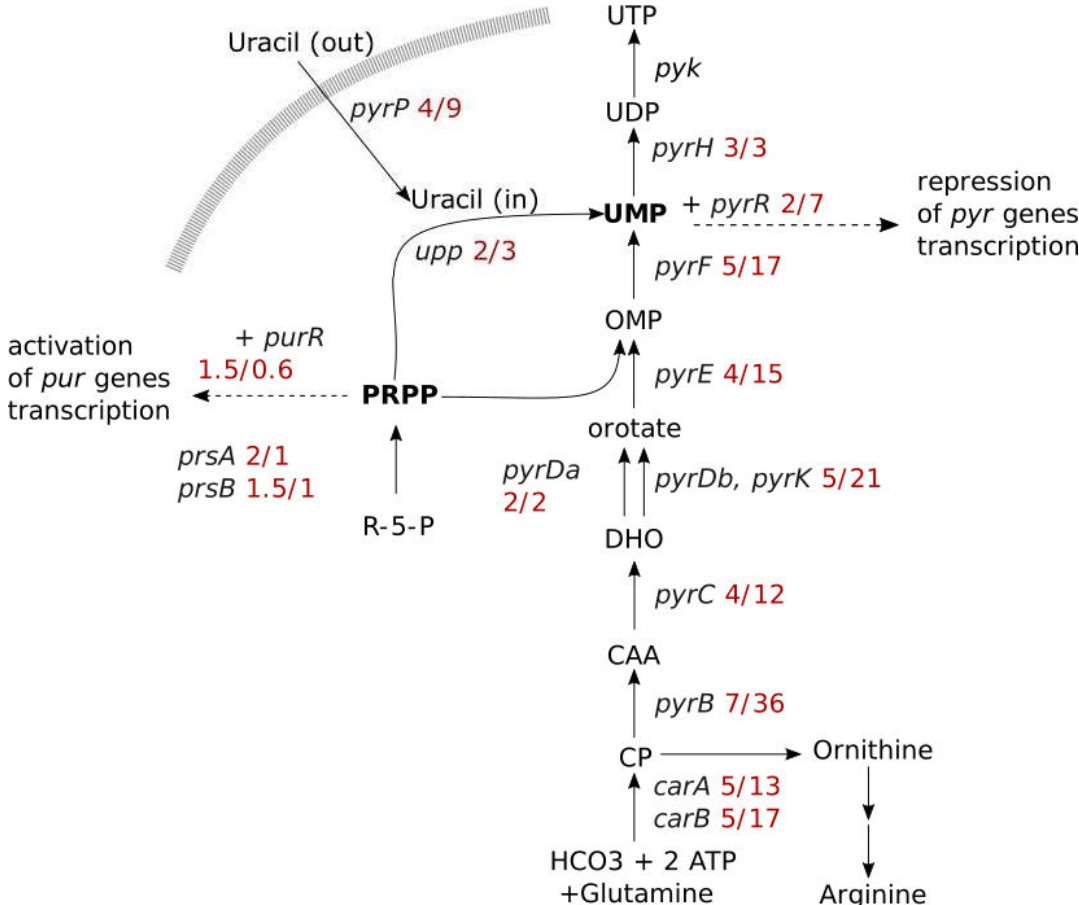

**FIG 3** Schematic of the UMP biosynthesis pathway. Genes are in italics for those that were downregulated; the fold change at 5 and 15 minutes is shown in red. The following abbreviations are used: CP, carbamoylphosphate; CAA, carbamoylaspartate; DHO, dihydroorotate; OMP, orotate monophosphate; *carAB*, carbamoylphosphate synthase; *pyrB*, aspartate transcarbamoylase; *pyrC*, dihydroorotase; *pyrDa*, dihydroorotate dehydrogenase A; *pyrDb/pyrK*, dihydroorotate dehydrogenase B; *pyrE,* orotate phosphoribosyltransferase; *pyrF*, OMP decarboxylase; *pyrH*, UMP kinase; *prsA*, *prsB*, PRPP synthase; *pyrR*, pyrimidine regulator; *pyrP*, uracil permease; *upp*, uracil phosphoribosyltransferase.

synthesis. Analysis with DeepTMHMM (44) suggests that the encoded protein possesses four transmembrane domains.

## Repression of other genes associated with nucleotide metabolism

We found that other genes associated with nucleotide metabolism were also repressed, though to a lesser extent (Table S1). Phosphoribosyl pyrophosphate (PRPP) is an essential molecule in the metabolism of both pyrimidine and purine nucleotides, as it is required for their *de novo* biosynthesis as well as for salvage of nucleobases (Fig. 3). The two genes encoding PRPP synthase, *prsA* and *prsB*, were both downregulated at 5 minutes p.i. in our study.

Ribonucleotides can be reduced to the corresponding deoxyribonucleotides by nucleotide reductases encoded by the *nrd* genes. *L. cremoris* has two classes of reductase genes found in two operons (*nrdHIEF* and *nrdDG*) (45). We found that five of the six *nrd* genes in these operons were repressed (Table S1).

Exogenous nucleobases and nucleosides can be utilized in salvage pathways following their uptake through the cell membrane by transport systems. In *L. cremoris*, the gene encoding uracil permease, *pyrP* (37), is a part of the *pyrRPBcarA* operon discussed above and was strongly downregulated. In addition, three of the four

genes encoding the BmpA-NupABC subunits of the correspondingly named nucleoside transport system (46) were downregulated (Table S1).

These findings revealed a shutdown of the *de novo* and salvage pathways for nucleotide production combined with reduced import of nucleobases and nucleosides.

## Upregulation of ribosome biogenesis and translational capacity

We observed upregulation of many genes associated with mRNA translation by 5 minutes p.i. The 316 host genes found to be upregulated include 40 of the 56 ribosome proteins and three of the four initiation factors (*infA*, *infB*, and *infC*; Table S1). Furthermore, genes associated with translation that were upregulated include *tsf* (translation elongation factor Ts), *der* (ribosome biogenesis GTPase), *rbfA* (30S ribosome-binding factor), *rimP* (ribosome maturation factor), and *rsfS* (ribosome silencing factor).

Some translation-associated genes were downregulated. These include *frr* (ribosome recycling factor), *raiA* (ribosome-associated translation inhibitor), *yihA* (ribosome biogenesis GTP-binding protein), *lepA* (translation elongation factor 4), and three ribosome proteins: *rpmE*, *rpmB*, and *rplO*. The regulation of both *lepA* and *raiA* is noteworthy given that both are known to increase the fidelity of translation (47, 48). The repression of these genes reveals a gene expression program that trades fidelity for capacity (49). The repression of *frr* which releases post-termination ribosomes from the mRNA was shown to trigger ribosome rescue mechanisms (50) and to affect the coupled translation of downstream ORFs (51). Nevertheless, we conclude that the regulation observed is likely to increase ribosome protein biogenesis and the total translational capacity of the cells.

## Regulation of chaperone systems

We observed the regulation of two major chaperone systems regulating protein folding. *dnaK* and *grpE* are both found to be upregulated, whereas the *groES/groEL2* are both repressed by 5 minutes p.i. In addition to repairing misfolded proteins (52), the chaperone DnaK and its nucleotide exchange factor GrpE are known to facilitate DNA replication of *E. coli* phage lambda (53, 54). Perhaps, an upregulation of this chaperone system might be similarly beneficial for sk1. Conversely, the observed downregulation of the GroES/GroEL chaperone system may have a negative effect on sk1 propagation, as it has been shown to aid with the folding of the coat protein of bacteriophages, e.g., P22 (55).

## Changes of ribosome silhouettes associated with infection

A comparison of the start codon metagene profiles from sk1- and mock-infected samples revealed a reduced density of ribosome footprints at the beginning of the coding region which was more pronounced in the sk1-infected samples (Fig. 4A). This run-off appearance would typically occur due to relative elongation to initiation rate increase. The cause for the apparent run-off is not clear but may indicate reduced initiation as well as increased elongation.

We further explored the distribution of ribosome footprints along individual transcripts (we term this a ribosome silhouette as it reflects the overall shape rather than the depth of ribosome coverage). Our approach consisted of comparing the similarity of ribosome silhouettes between replicates to that between states, i.e., sk1 or mock infected (36) (see Materials and Methods).

This revealed that indeed the ribosome silhouette differs reproducibly for most genes in mock control vs sk1-infected samples (Fig. 4B). An example is provided of a pronounced change in the profile of LLNZ_RS07660 which encodes a glycosyltransferase family 39 protein (Fig. 4C).

A pairwise comparison of ribosome silhouettes across the time course (2 minutes mock control, 2 minutes sk1 infected, and 15 minutes sk1 infected; Fig. S6) enabled us to get a better understanding of the temporal aspect of observed changes in a given

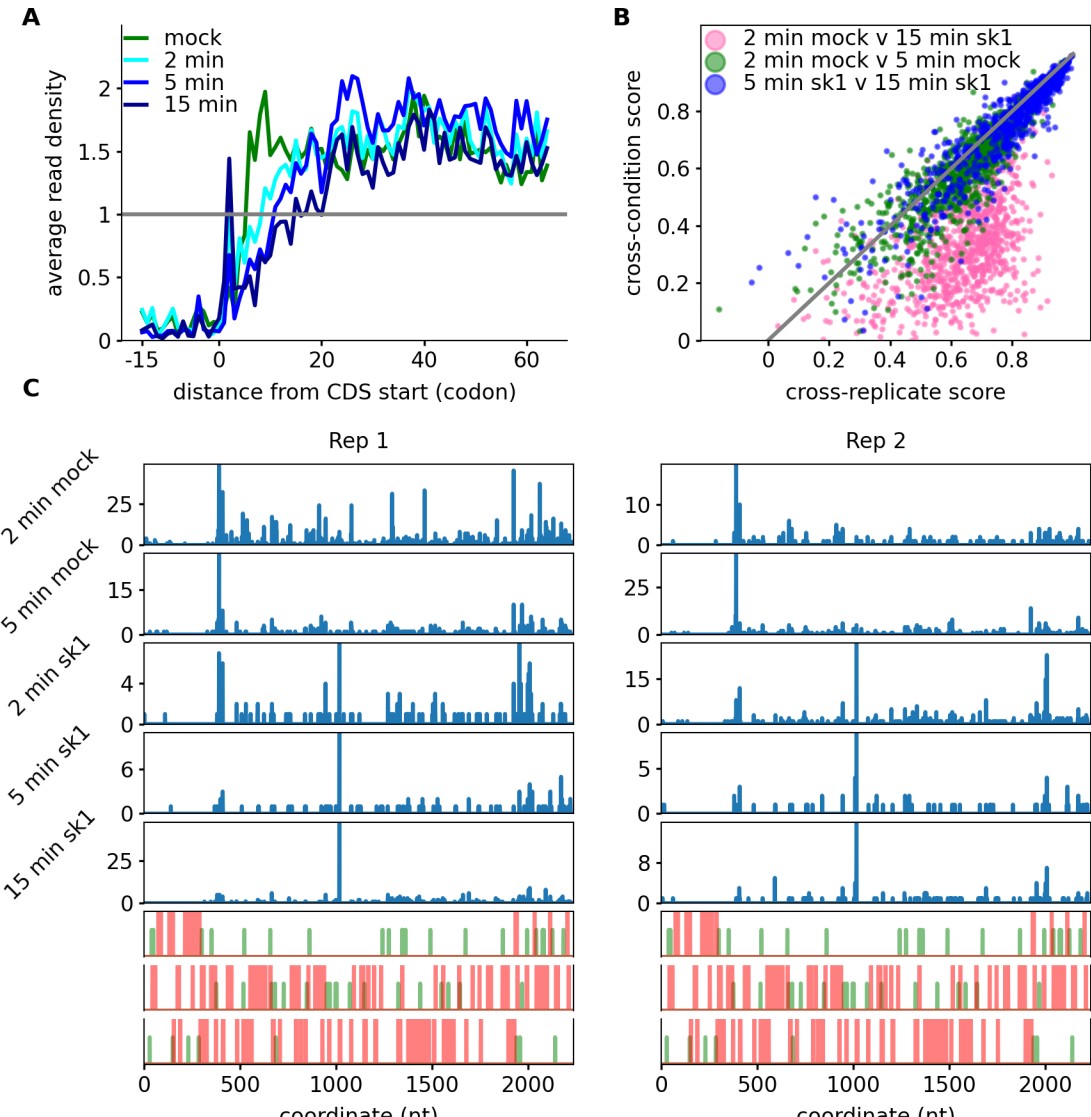

**FIG 4** Changes in ribosome silhouettes with infection. (A) Metagene profiles at the vicinity of the start codon for mock- and sk1-infected samples. (B) Comparison of ribosome silhouettes using the cross-replicate score (median Spearman's correlation coefficient between replicates, *x*-axis, between conditions, or *y*-axis). Note the deviation from the diagonal of individual gene data points for the "2 minutes mock vs 15 minutes sk1." (C) Ribo-seq profiles of LLNZ_RS07660 provide an example of differential silhouettes between mock and sk1 samples. Both independent replicates of each sample are shown in two adjacent panels. The number of mapped reads is indicated on the *y*-axis. The bottom panel contains an ORF organization plot, where green lines indicate ATG and red lines indicate stop codons. Observe how the profiles differ between mock- and sk1-infected samples but not between replicates (of the same time points).

silhouette. We found that the ribosome silhouettes of the 2-minute p.i. are more similar to that of the 15-minute p.i. time point when compared to those of the mock-infected control samples. Indeed, the difference in silhouettes of the 2 and the 15-minute sk1 infection time points is comparatively minor. Thus, the change in ribosome silhouettes occurs principally within 2 minutes p.i. These changes in silhouettes may indicate changes in translation rates upon infection.

## Bimodal distribution of leader length in *L. cremoris*

We applied end-enriched RNA-sequencing (Rend-seq) (56) to capture the original 5′ transcript ends. This approach relies on mild RNA digestion resulting in the overrepresentation of footprints containing the original 5′ or 3′ mRNA ends. This is because the

original transcript ends need not be digested to become ends of a footprint, unlike the internal transcript positions which can only become footprints ends upon cleavage (56).

With these data, we identified 260 putative transcription start sites (TSS). Two hundred twenty of the putative TSS were also identified as such in a closely related *L. cremoris* strain (MG1363) (8). Consistent with the data from the latter study, the read length distribution appears to be bimodal (Fig. 5A). The first mode consists of genes with a leader length of less than five nucleotides, and the second mode consists of genes with leaders of approximately 40 nucleotides in length (Fig. 5A). We did not identify any leaders to be within 4 and 14 nucleotides in length. It seems likely that leaders of this length range are problematic for ribosome initiation, which likely points to mechanistic differences in initiation at leaderless and leadered mRNAs (Fig. 5A). A Rend-seq metagene profile of TSS of genes classified to be leaderless shows the

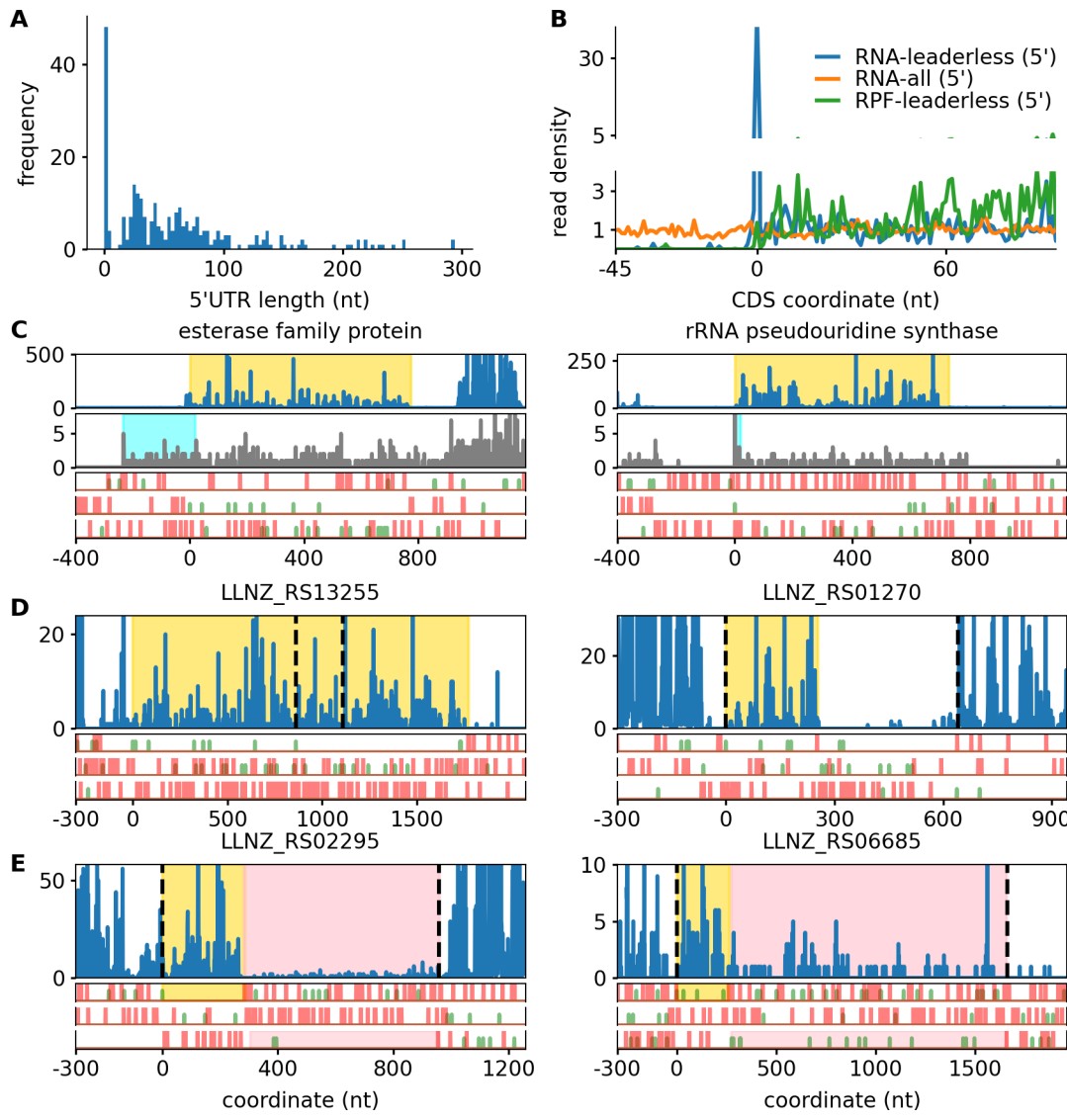

**FIG 5** Toward an improved annotation of *L. cremoris.* (A) Length distribution of the identified 5′ leaders. (B) Metagene profiles at the start codon of genes are identified to be leaderless. Note the major peak density in blue (30× compared to the background) at the 0 coordinate. (C) Ribo-seq (blue) and RNA-seq (gray) profiles for LLNZ_RS09750 (encoding esterase family protein; left) and LLNZ_RS12635 (encoding rRNA pseudouridine synthase; right) for both of which TSS were identified in this study. The 5′ leaders are highlighted in cyan. (D) Ribo-seq profiles of two genes previously annotated as pseudogenes, now found to be translated. (E) Ribo-seq profiles of two genes previously annotated as pseudogenes, that contain two translated ORFs. Dotted lines indicate the current annotation, and newly annotated ORFs are highlighted in yellow and pink.

expected enrichment, with peak density 30× that of average RNA density (Fig. 5B). Two examples of newly identified TSS are provided (Fig. 5C).

## Translation of 60 annotated pseudogenes

Pseudogenes are regions of DNA that resemble genes but are expected to be nonfunctional. They are typically annotated as such because of the presence of an indel or nonsense mutation in a region displaying high sequence similarity to a known coding region. We examined the alignments to pseudogenes and identified that 60 of the 155 genes annotated as pseudogenes contain translated ORFs (see Materials and methods). Nineteen of these genes were annotated independently as coding during this manuscript preparation. For 53 of the 60 genes we annotated, translation occurs at the ancestral CDS start site. For 49 cases, the length of the translated ORF was truncated compared to that of the corresponding functional gene (Table S1). Most pseudogenes exhibited low gene expression, with the expression of only eight genes exceeding the median (Table S1). Two examples of translated pseudogenes are provided (Fig. 5D).

Among the 60 cases, we found two genes with aligned footprints in two ORFs that may be owing to recoding events. LLNZ_RS06685 encodes a glycosyl hydrolase and was found to be a candidate of transcriptional slippage (Fig. 5E). This gene was found to have a deletion of a G at a TAA_AAA_AAA_**G**AC_AAA_ATC (where underscores indicate the original reading frame) relative to most other strains, which produces the TAA_AAA_AAA_ACA_AAA_TC. A −1/+2 shift is required to restore the original reading frame and express the full-length protein. The highly repetitive use of 12 A-T pairs in a 13-nucleotide DNA region could enable a shift of the complementary RNA during transcription. The efficiency of translation of the full-length product relative to the short product was estimated to be 18% based on the ribosome profile.

LLNZ_RS02295 was found to be a candidate of ribosomal frameshifting (Fig. 5E). The gene was found to have a deletion, relative to most other strains, of a C at AAT_TT**C**_TTT_TTA_A and resulting in the AAT_TTT_TTT_TAA sequence. This contains the classical −1 frameshifting slippery X_XXY_YYZ sequence motif (57), and indeed, a −1/+2 frameshift is required to produce the full-length protein product. We could not exclude the possibility that the shift is due to transcriptional slippage in this case too. The efficiency of translation past the putative shift site was estimated to be 6%. The encoded protein is described as a CdaR family protein. The indel has evolved recently as a nucleotide BLAST query revealed that it was only observed in one other strain (*L. cremoris* MG1363).

## DISCUSSION

Our study revealed a profound alteration in the nucleotide metabolism of *L. cremoris* upon sk1 phage infection. The shutdown of nucleotide import and salvage pathways coupled with the downregulation of various *pyr* operons suggests that the primary aim of this response is to restrict the availability of nucleotides. The results are consistent with the finding that *L. cremoris* mutants that are unable to produce dTTP are resistant to phage infection (58). While further experiments are needed to confirm that this is an antiphage defense mechanism aimed at preventing phage DNA replication or inducing self-sacrifice, we consider it the most plausible explanation for our findings in light of the growing body of evidence supporting nucleotide limitation as a universal antiviral response (59–62). For example, human cells have been shown to utilize this strategy, where the protein SAMHD1 hydrolyzes dNTPs to deplete the cellular dNTP pool and restrict HIV replication (61, 62).

Recently, nucleotide limitation has been identified as a widespread antiphage mechanism in the proteobacteria phylum (59, 60). This host response is mediated by the activity of deoxycytidylate deaminase enzymes that metabolize DNA nucleotides upon infection, thereby limiting the building blocks for phage genome replication. The deaminated dCTP and dCMP are further converted to dUMP, resulting in increased cellular UMP levels. It is conceivable that a similar activity occurs in *L. cremoris*, where

an increase in UMP levels upon phage infection could result in PyrR-UMP-mediated downregulation of *pyr* operons. This could be an effective approach that integrates the breakdown of dNTPs to UMP with subsequent UMP-mediated inhibition of *de novo* nucleotide synthesis, quickly depleting the cell of dNTPs.

While nucleotide limitation is likely to benefit the host, the upregulation of genes associated with protein synthesis and chaperones may benefit phage propagation. Given the significant and rapid nature of these regulatory changes, it is plausible that the observed regulation is actively induced by the phage. Furthermore, the repression of genes associated with translational fidelity suggests a trade-off between accuracy and speed. This might be a successful tactic if the phage exhibits a higher tolerance for translational errors compared to its host.

Clarifying the complex response arising from the conflicting interests of the phage and its host could provide valuable insights into the co-evolution of phage–host interactions (63). Despite the significant suppression of nucleotide production, sk1 phage infection remains highly virulent. This suggests that sk1 has evolved mechanisms to bypass this defense strategy. The rapid and pronounced downregulation of *pyr* genes also supports the notion of antiviral response, as such a swift and profound reaction is unlikely to be part of normal homeostatic regulation. If induced by the phage, it would be a vulnerability for the host, which it would not retain during evolution.

We have generated the first ribosome profiling data from *L. cremoris*, which has allowed us to validate previous genome annotations that were based solely on comparative sequence analysis. Our analysis revealed that 60 loci annotated as pseudogenes are translated. While this may seem remarkable, a recent study reported that 101 out of 161 pseudogenes are translated into *Salmonella enterica* (64). These were mostly translated via alternative genetic decoding such as −1 frameshifting (65), illustrating the concept of "pseudo-pseudogenes" (64) when a seemingly deleterious mutation such as an indel is compensated by shift or slippery-prone sequence patterns. In our study, we identified only two putative cases of alternative genetic decoding and instead found that most translated pseudogenes encode truncated proteins with a median loss of about 50% of the original protein sequence. Whether these truncated protein products retain some of the ancestors' functions or whether they acquired novel biological functions or became toxic remains to be investigated.

Our application of Ribo-seq to investigate gene expression during phage infection highlights the method's potential to precisely capture alterations in gene expression during early and intermediate stages of phage infection, thereby arguing for its wider adoption within the field of phage–host interactions.

## MATERIALS AND METHODS

### *L. cremoris* growth, infection course, and harvest

For each time interval, there were two sk1- and two mock-infected replicates. For each, 200 mL of M17 broth supplemented with 5 g/L glucose and 10 mM $CaCl_2$ were inoculated with overnight culture to $OD_{600nm}$ of ~0.05 and incubated at 30°C without agitation to an $OD_{600nm}$ of 0.4–0.43. To achieve a multiplicity of infection of ~10, 12 mL of phage sk1 in SM buffer (100 mM NaCl, 50 mM Tris HCl, and 10 mM $MgSO_4$) with a titer of $5 \times 10^{10}$ per mL were added to the two test samples. The same volume of SM buffer alone was added to the two mock-infected cultures at the same time. At 2, 5, and 15 minutes p.i., to stall translating ribosomes, chloramphenicol (100 µg/mL) was added for 2 minutes prior to harvesting. The cell cultures were then poured into centrifuge tubes filled with equal volume of crushed ice supplemented with 100 µg/mL chloramphenicol and centrifuged at 8000 rpm (10,000 × g) for 2 minutes at 4°C to harvest the cells. The pellets were resuspended in 2 mL of ice-cold lysis buffer [10 mM $MgCl_2$, 100 mM $NH_4Cl$, 20 mM Tris pH 8.0, 0.1% NP-40, 0.4% Triton X-100, 1 mM chloramphenicol, and 100 U/µL DNase I (Sigma-Aldrich)] and pipetted dropwise into 50 mL falcon tubes filled 1/3 with liquid nitrogen.

## Ribosome profiling

Frozen droplets were pulverized in prechilled 10 mL grinding jars in Retsch Mixer Mill 400 for seven cycles each 1.5 minutes at 21 Hz. Jars were chilled in liquid nitrogen between cycles. Pulverized lysates were thawed on ice and clarified by centrifugation at 20,000 g for 10 minutes at 4°C. Twenty-five AU of lysate was supplemented with 5 mM $CaCl_2$ and digested with 1500 U of Micrococcal nuclease (Roche) at 25°C for 1 hour with agitation at 1400 rpm. The reaction was quenched by 6 mM EGTA. The digested lysates were loaded onto 10%:50% manually prepared sucrose gradients and centrifuged in an SW41-Ti rotor (Beckman Coulter) at 40,000 rpm for 3 hours at 4°C. Brandel Density Gradient Fractionator was used to isolate monosome fractions at 1.5 mL/minute flow speed. RNA was extracted with Trizol (Invitrogen) from the collected monosome fractions. Ten micrograms of monosome-derived RNA was loaded on a 15% TBE-Urea polyacrylamide (PAA) gel, and footprints with sizes ranging from 15 to 45 nucleotides were excised from the gel.

## Library construction

The extracted footprints were treated with 10 U T4 Polynucleotide Kinase [New England Biolabs (NEB)] for 1 hour at 37°C to prepare the 3′ ends for ligation with a DNA linker. T4 RNA Ligase 2, truncated K227Q (NEB) was used to ligate footprints with pre-adenylated linkers bearing unique molecular identifiers (UMIs) and sample barcodes derived from reference (54). Ligation products were excised from a 10% TBE-Urea PAA gel, and samples with different barcodes were pooled together. These were subjected to rRNA removal with Ribo-Zero Magnetic kit for Gram-positive bacteria (Epicenter) according to the manufacturer's protocol. cDNA was generated using SuperScript III Reverse Transcriptase (Invitrogen) for 30 minutes at 48°C and isolated by size excision on a 10% TBE-Urea PAA. Single-stranded cDNA was circularized using 100 U of CircLigase (Epicenter) for 1 hour at 60°C. Circularized ribosomal footprint cDNA was amplified by PCR using the Phusion High-Fidelity enzyme (NEB). After four rounds of PCR amplification, the product was recovered by size excision from an 8% TBE PAA gel. Deep sequencing was performed with the Illumina HiSEQ4000. All primers used, such as the pre-adenylated linkers, reverse transcription primers, and indexed PCR primers, were derived from reference (66).

## RNA-seq and Rend-seq

Total RNA was extracted from cell lysates using Trizol reagent (Invitrogen). Ten micrograms of total RNA was subjected to alkaline fragmentation in the presence of 50 mM $NaHCO_3$, 50 mM $Na_2CO_3$, and 0.25 M EDTA for 20 minutes at 95°C. The reaction was stopped with 320 mM NaOAc, pH 5. Following isopropanol precipitation, fragments in the 20–60 nt size range were obtained by excision from a 15% TBE-Urea PAA gel. Library construction was performed as described for ribosome profiling. The same procedure was followed for REND-seq except that zinc-mediated fragmentation was performed for 65 seconds at 95°C as in reference (56).

## Initial data processing

The adapter sequence (AGATCGGAAGAGCACACGTCTGAA) was removed by Cutadapt. Next, reads were binned based on the 5-nucleotide barcode sequence at the 3′-end of the reads with a custom script. The remaining sequence consisted of the ribosome footprint and a UMI sequence (the first two nucleotides and the last five nucleotides). As multiple copies of these reads were likely the result of PCR duplicates, only one instance of each sequence was carried forward. The UMI was removed, and reads less than five nucleotides in length were discarded. Reads that mapped to rRNA or tRNA with Bowtie (version 1.2.3) were discarded.

## Quantification of gene expression and differential gene expression analysis

The remaining reads were mapped to the genomes of *L. cremoris* NZ9000 ( NC_017949.1) and sk1 (AF011378.1) with Bowtie. The annotation of *L. cremoris* was obtained from NCBI ftp site. When these files were obtained, they were last modified on 11 September 2021.

Differential gene expression analysis was performed with DESeq2 with a workflow described in reference (67). We had to discard the 15-minute mock-infected samples because they were found to have anomalies in the comparison of the replicates (Fig. S1). The metagenes were produced with coding genes of *L. cremoris* distant from another coding region by at least 50 nucleotides to prevent adjacent ORFs from influencing the metagene profile. In addition, the profiles included data from transcripts with greater than 10 mapped reads. The profiles of each transcript were normalized individually before incorporation into the average.

## Examination of changes in ribosome profile silhouette

Due to limited sequencing depth, certain genes may exhibit regions without mapped reads, leading to potential inaccuracies in similarity scores. The low number of mapped reads can result in artificially strong similarity scores, as many transcript coordinates may lack mapped alignments, creating an illusory agreement. To address this issue, we employed a compression technique on individual gene silhouettes, removing coordinates that lacked mapped reads in both silhouettes. The compressed silhouettes were then evaluated for similarity using Spearman's correlation coefficient. To ensure reliable comparisons, each transcript profile required a minimum of 50 mapped reads across 10 distinct positions. To determine if gene silhouettes differed between two states (e.g., 2 minutes mock- vs 2 minutes sk1-infected), we compared the median pairwise similarity within replicates to that between states.

## Identification of transcriptional start sites

The identification of TSS was carried out on aggregated Rend-seq and RNA-seq data derived from the mock-infected samples at the 2 and 15 minutes p.i. time points. An initial sweep of the genome alignments identified coordinates that had a minimum of five mapped reads and an average read density of 0.1 reads per nucleotide across a 200 nt window (from −100 to +100).

For each position, we determined two measurements, the peak density and the change of density. The "peak density" at a genomic coordinate is the size peak relative to the average RNA-seq density in a 200 nt window. The "delta density" at a genomic coordinate is the increase of average RNA-seq density in a 100 nt upstream window to the density 100 nt downstream.

$$Peak\_density_n = \frac{RNA_n}{\frac{\sum_{m=-100}^{m100} RNA_{(n+m)}}{200}}$$

$$Delta\_density_n = \frac{\sum_{m=-100}^{m=0} RNA_{(n+m)} + 0.001}{\sum_{m=1}^{m=100} RNA_{(n+m)} + 0.001}$$

An examination of the two variables (peak density and delta density) enabled good discrimination between positions previously identified as TSS and those annotated as coding (used as true negatives) (Fig. S7). This approach was used to identify 325 candidate TSS. These were human evaluated which resulted in 260 candidates. Candidates were classified as being common to that reported by reference (8) based on the identical sequence similarity in the first 30 nucleotides of each gene.

## Identification of translated pseudogenes

To identify novel genes, our approach involved initially identifying genes with nonstandard characteristics in the current annotation. These characteristics included the absence

of ATG as the first codon, the absence of a stop codon (UAG, UAA, and UGA) as the last codon, or the presence of a stop codon within the annotated coding region. Subsequently, these genes were ranked based on the total number of aligned footprints, and the profiles were subjected to human evaluation. The positions of the start and stop codons were selected based on their ability to best explain the gene profile.

## ACKNOWLEDGMENTS

We gratefully acknowledge Darren Fenton for his invaluable assistance in harvesting samples at precise time intervals. We would also like to express our appreciation to Philip Kelleher and Joanna Kaczorowska for their valuable contributions in procuring reagents and facilitating the experimental work, respectively.

The research was conducted with the financial support of Science Foundation Ireland to D.v.S. [13/IA/1953 and 12/RC/2273-P2], and to J.M. [20/FFP-P/8664], and also SFI-HRB-Wellcome Trust Biomedical Research Partnership (Investigator in Science award) to P.V.B. [210692/Z/18].

## AUTHOR AFFILIATIONS

[1]School of Biochemistry and Cell Biology, University College Cork, Cork, Ireland
[2]EIRNA Bio, Bioinnovation Hub, Cork, Ireland
[3]School of Microbiology and APC Microbiome Ireland, University College Cork, Cork, Ireland

## AUTHOR ORCIDs

Douwe van Sinderen ⓘ http://orcid.org/0000-0003-1823-7957
Martina M. Yordanova ⓘ http://orcid.org/0000-0001-9693-3857

## FUNDING

| Funder | Grant(s) | Author(s) |
| --- | --- | --- |
| Science Foundation Ireland (SFI) | 13/IA/1953, 12/RC/2273-P2 | Douwe van Sinderen |
| Science Foundation Ireland (SFI) | 20/FFP-P/8664 | Jennifer Mahony |
| Wellcome Trust (WT) | 210692/Z/18 | Pavel V. Baranov |

## AUTHOR CONTRIBUTIONS

Patrick B. F. O'Connor, Investigation, Methodology, Validation, Visualization, Writing – original draft | Jennifer Mahony, Conceptualization, Funding acquisition, Project administration, Resources, Supervision, Writing – review and editing | Eoghan Casey, Methodology | Pavel V. Baranov, Conceptualization, Funding acquisition, Supervision, Writing – review and editing | Douwe van Sinderen, Conceptualization, Funding acquisition, Project administration, Resources, Supervision, Writing – review and editing | Martina M. Yordanova, Conceptualization, Investigation, Methodology, Supervision, Validation, Visualization, Writing – original draft, Writing – review and editing

## DATA AVAILABILITY

The sequences of the ribosome profiling libraries have been deposited in the NCBI Gene Expression Omnibus portal under the accession number GSE231968.

## ADDITIONAL FILES

The following material is available online.

## Supplemental Material

**Fig. S1 (Spectrum03989-23-s0001.tiff).** Comparison of the number of mapped reads per gene across replicates.

**Fig. S2 (Spectrum03989-23-s0002.tiff).** Subcodon periodicity signal distribution obtained across all Ribo-seq and RNA-seq samples.

**Fig. S3 (Spectrum03989-23-s0003.tiff).** New translated ORF found in sk1.

**Fig. S4 (Spectrum03989-23-s0004.tiff).** Gene expression response consists of a major shift within 2 minutes p.i.

**Fig. S5 (Spectrum03989-23-s0005.tiff).** LLNZ_RS05640/orfC confirmed to belong to the *pyrKDbForfC* operon.

**Fig. S6 (Spectrum03989-23-s0006.tiff).** Changes in ribosome profile silhouettes occur mostly within 2 minutes p.i.

**Fig. S7 (Spectrum03989-23-s0007.tiff).** Identification of transcriptional start sites.

**Supplemental legends (Spectrum03989-23-s0008.docx).** Legends for Fig. S1 to S7.

**Table S1 (Spectrum03989-23-s0009.xls).** Results of sequencing data analysis tabulated in several sheets.

## Open Peer Review

**PEER REVIEW HISTORY (review-history.pdf).** An accounting of the reviewer comments and feedback.

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
