## [Reviewer comments · Microbiology Spectrum]

Microbiology Spectrum

Ribosome profiling reveals downregulation of UMP biosynthesis as the major early response to phage infection.

Patrick O'connor, Jennifer Mahony, Eoghan Casey, Pavel Baranov, Douwe van Sinderen, and Martina Yordanova

Corresponding Author(s): Martina Yordanova, University College Cork Department of Biochemistry and Cell Biology

Review Timeline:

Submission Date:	November 23, 2023
Editorial Decision:	January 29, 2024
Revision Received:	February 6, 2024
Accepted:	February 14, 2024

Editor: Thomas Denes

Reviewer(s): The reviewers have opted to remain anonymous.

Transaction Report:

DOI: <https://doi.org/10.1128/spectrum.03989-23>

Re: Spectrum03989-23 (Ribosome profiling reveals downregulation of UMP biosynthesis as the major early response to phage infection.)

Dear Dr. Martina M Yordanova:

We received one review with some minor suggestions. I have also read through your manuscript and agree that the article can be quickly accepted with these minor modifications. Below you will find my comments, instructions from the Spectrum editorial office, and the reviewer comments.

Revision Guidelines

Sincerely,
Thomas Denes
Editor
Microbiology Spectrum

Reviewer #1 (Comments for the Author):

The paper of O'Connor et al. describes the use of Ribo-seq and RNA-seq for analyzing the response of the *Lactococcus cremoris* translatoome and transcriptome to infection with the sk1 phage.

Critique,

It is an important and technically sound survey of alterations in expression of the host and phage genes within few minutes post-infection. The authors make an excellent job in trying to make sense of the massive data with hundreds of genes undergoing up- or down-regulation. The most consistent pattern, the downregulation of genes involved in UMP biosynthesis, seems to be of a high relevance to the host-phage interaction. Whether such a response benefits the phage or the host remains to be examined, but pinpointing it is already an important achievement. The other cellular responses are more ambiguous. The upregulation of some ribosomal protein genes is accompanied with the downregulation of some other ribosomal protein genes, a more active expression of some chaperones occurs simultaneously with the downshift in expression of the others. Regardless, the discussion of such effects is accurate, insightful and thought provoking.

The paper is well written. It is easy to follow the authors' train of thoughts and the discussion is nicely balanced and insightful.

I have very few suggestions how to further improve the manuscript:

II. 187-197. Since PyrR controls transcription of the dramatically downregulated pyr operon, it would be helpful if the authors commented on PyrR expression. It would be also good if they discuss whether all the changes of the multiple genes of the UMP biosynthesis could be attributed to the altered function or expression of a single regulatory protein.

Fig. 1B. Define red dots.

Fig. 1D: It is unclear what distinguishes "unambiguous footprints" from "all".

Reviewer' s comments

The paper of O'Connor et al. describes the use of Ribo-seq and RNA-seq for analyzing the response of the *Lactococcus cremoris* translome and transcriptome to infection with the sk1 phage.

Critique,

It is an important and technically sound survey of alterations in expression of the host and phage gene within few minutes post-infection. The authors make an excellent job in trying to make sense of the massive data with hundreds of genes undergoing up- or down-regulation. The most consistent pattern, the downregulation of genes involved in UMP biosynthesis, seems to be of a high relevance to the host-phage interaction. Whether such a response benefits the phage or the host remains to be examined, but pinpointing it is already an important achievement. The other cellular responses are more ambiguous. The upregulation of some ribosomal protein genes is accompanied with the downregulation of some other ribosomal protein genes, a more active expression of some chaperones occurs simultaneously with the downshift in expression of the others. Regardless, the discussion of such effects is accurate, insightful and thought provoking.

The paper is well written. It is easy to follow the authors' train of thoughts and the discussion is nicely balanced and insightful.

****Response****

We are grateful for the reviewer's favourable comments.

I have very few suggestions how to further improve the manuscript:

II. 187-197. Since PyrR controls transcription of the dramatically downregulated *pyr* operon, it would be helpful if the authors commented on PyrR expression. It would be also good if they discuss whether all the changes of the multiple genes of the UMP biosynthesis could be attributed to the altered function or expression of a single regulatory protein.

****Response****

*The protein PyrR is a major regulator of *pyr* genes expression. Upon UMP binding, the protein PyrR-UMP complex interacts with *pyr* mRNA leaders, allowing the formation of a transcription termination structure and downregulating expression of *pyr* genes. Since PyrR is a master regulator of *pyr* genes expression and because our study showed such a pronounced and coordinated downregulation of *pyr* mRNA translation, we consider it plausible for PyrR to be involved in the observed effect.*

*To clarify PyrR's possible role, we made a change in the text in Discussion (343-346). We changed the following sentence: "It is conceivable that a similar activity occurs in *L. cremoris*, where an increase in UMP levels upon phage infection would lead to the coordinated downregulation of *pyr* operons observed in our study." to "It is conceivable that a similar activity occurs in *L. cremoris*, where an increase in UMP levels upon phage infection could result in PyrR-UMP mediated downregulation of *pyr* operons."*

However, we have no direct evidence for PyrR's role and we couldn't make conclusions based on its expression levels. pyrR itself is a component of one of the pyr operons, and its expression levels are similarly reduced. However it is plausible that preexisting PyrR protein binds to UMP during infection, particularly if UMP levels rise, as indicated by recent studies. Under such conditions PyrR might even experience stabilization upon UMP binding.

We deliberately refrain from placing too much emphasis on the speculative role of PyrR, as our current study lacks solid evidence of its involvement. We hope that future studies will shed further light on this matter.

Fig. 1B. Define red dots.

&

Fig. 1D: It is unclear what distinguishes "unambiguous footprints" from "all".

*****Response*****

We are grateful to the reviewer for noticing these.

We updated Figure 1B legend to include: "Red dots indicate sk1 genes".

We also updated Figure 1D legend to include: "Unambiguous footprint reads (in red) are those which mapped only to single position in the genome while "all" (in blue) includes those reads that mapped to multiple positions. RNA-seq reads are in green." We now also define "unambiguous reads" in the main text.

Re: Spectrum03989-23R1 (Ribosome profiling reveals downregulation of UMP biosynthesis as the major early response to phage infection.)

Dear Dr. Martina M Yordanova:

Your manuscript has been accepted, and I am forwarding it to the ASM production staff for publication. Your paper will first be checked to make sure all elements meet the technical requirements. ASM staff will contact you if anything needs to be revised before copyediting and production can begin. Otherwise, you will be notified when your proofs are ready to be viewed.

Sincerely,
Thomas Denes
Editor
Microbiology Spectrum